# ANN-Based Estimation of the Defect Severity in the Drilling of GFRP/Ti Multilayered Composite Structure

**Igor Zhilyaev [1,\*], Evgeny Chigrinets [2], Sergey Shevtsov [3,\*] , Samira Chotchaeva [2] and Natalia Snezhina [2]**

1   Institute of Polymer Engineering, The University of Applied Sciences Northwestern Switzerland FHNW, CH–5210 Windisch, Switzerland
2   Department. of Aircraft Engineering, Don State Technical University, 344002 Rostov on Don, Russia
3   Department. of Transport, Composite Materials and Structures, Southern Center of Russian Academy of Science, 344006 Rostov on Don, Russia
\*   Correspondence: igor.zhilyaev@fhnw.ch (I.Z.); sergnshevtsov@gmail.com (S.S.)

**Abstract:** The main purpose of this study was to develop a model for predicting the quality of holes drilled in the root part of the spar of helicopter main rotor blades made of glass fiber-reinforced plastic (GFRP)-Ti multilayer polymer composite. As the main quality criterion, delaminations at the entry and exit of the drill from the hole were taken. In the experimental study, a conventional drill and two modified geometry drills, a double-point angle drill and a dagger drill, were used. Preliminary experiments showed the best hole quality when using modified drills, which allowed further detailed study only with both modified drills at different drilling speeds and feed rates. Its results in the form of training sets were used to build artificial neural networks (ANNs) to predict delamination at the entry and exit of the drilled holes. An analysis of the fitted response functions presented as 3D surface plots and contour plots led to the selection of the best tool, a double-point angle drill, which demonstrated the lowest achievable delamination both at the entry and at the exit of the holes approximately 1.5 times less (0.45/0.48 mm) compared to dagger drills (0.68/0.7 mm) and determined the ~5 times larger optimal area for the drilling speed and feed rate. The results obtained confirm the possibility of effective prediction of the quality and productivity of mechanically processed composites of complex reinforcement using ANN to quantify the quality criteria and search for the optimal modes of such technologies.

**Keywords:** polymeric composite; GFRP/Ti reinforcement; hole drilling; delaminations; defect prediction; tool geometry; drilling modes

## 1. Introduction

A large number of particularly critical aircraft structures made of polymer composite materials are subjected to machining, in particular, drilling of holes for fastening the composite structure to other structural elements of the aircraft. Such holes are made in the blades of the main and tail helicopter rotors for fastening to the hub in the structures of stabilizers, wings, and various parts of the fuselages. The necessity and difficulties of fulfilling the increased requirements for the quality of such holes were considered in detail in the works [1–4]. The difficulty of achieving the quality of machining holes in high-strength polymer composites, especially those reinforced with metal layers, is mainly due to the fact that they are multi-component discrete materials with different mechanical, abrasive and thermal properties of the components. Studies [2–9] summarize the following features of polymeric composites during their machining:

-   Low thermal conductivity and high hardness of reinforcing glass fibers, leading to wear and blunting of the cutting edges of the tool.
-   The dependence of the rheological state (viscosity) of both thermosetting and thermoplastic matrices on the temperature during drilling makes it difficult to remove

chips from the hole and degrade the properties of the resin when heated. Moreover, the adhesion of the resin to the metal layers is significantly reduced as a result of its heating due to the heat generated during cutting.

- Moisture absorption, inherent in most polymer composites, does not allow the use of cooling fluids during drilling to reduce the temperature and degradation of resins.
- A sharp difference in the mechanical properties and machinability of titanium and polymer composite causes crushing of thin interlayer titanium plates near the surface of the hole to be machined, contributing to the emergence of the splintering or fiber breakout, fiber pullouts, and delaminations.

High-quality processing of holes in carbon/glass fiber-reinforced plastic (CFRP/GFRP)/ metal stacks, which is a sequence of alternating layers of composite and metal, is the most difficult, and new varieties of such materials being created require a specific approach to the analysis and development of appropriate technologies. It was experimentally established in [10–16] that when the cutting edge of the drill moves from the composite to the titanium layer, there is an abrupt increase in temperature, which causes melting of the polymer matrix, as well as a sharp growth in the thrust force, which is most dependent on the feed rate during drilling [3–6,8,15,17]. The authors of [5,8,15,17,18] also report the occurrence of significant delaminations at the interface between two dissimilar materials.

The undesirable temperature effects and delaminations that occur during drilling of such composites are the main reasons for the degradation of the structural strength of aircraft structures manufactured using these materials [5,6,10,19–21]. A thorough analysis of the mechanisms heat generation during drilling of reinforced polymer composites [1,16,22], extensive experimental results of thermal effects presented in articles [11,21–25], and studies of related degradation processes of the polymer matrices properties [26–28] have been  performed.

The occurrence and assessment of the material properties' degradation as a result of the formation of delaminations during drilling, which adversely affect both the short-term and long-term strength of composite structures, have been studied in many experimental works, such as [10,12,20,25]. Their results strongly suggest that delamination is most likely to occur at the points of entry and exit of the drill from the hole being machined. There are also studies aimed at finding a quantitative relationship between the magnitude of unacceptable splintering or fiber breakout, fiber pullout, delaminations, the place of their formation, tool geometry, and drilling modes [5,6,8,15]. The monographs [5,6] present the results of the influence of fibers orientation and lay-up sequence on the tendency of the composite to delamination during drilling. They show that angle-ply laminates provide better-machined surfaces than unidirectional laminates. The best surface quality is obtained when fibers are sheared off at a right angle (90°), while the worst surface is obtained when fibers are compressed and bent, which occurs at intermediate angles (20° to 45°).

Studies of the conditions for the occurrence and development of the described defects using the most modern experimental equipment, including optical and scanning electron microscopy (SEM), Raman and Fourier-transform infrared (FTIR) spectroscopy; reconstruction of the temperature field by using high sensitive thermocouples integrated inside the specimens with multi-sensorial data acquisition system, thermal analysis, and high precision mechanical testing according to the required ASTM standards, are presented in works [11,16,29–39]. Their results made it possible to study the thermal effects in polymer matrices in detail, to propose methods for reducing their influence by introducing some fillers, and to reduce unwanted deformations due to the rational choice of process modes. Numerous studies have also confirmed that hole quality improvement is achieved at low feed rates and cutting speeds increased to a certain limit, above which the heating effect becomes significant, especially when drilling stacked glass/carbon fiber and metal. The experience of machining holes gained in manufacturing has been the source of process improvements such as the use of the back support of the hole to prevent delamination at the drill exit; peck drilling, in which the drill advances into material, retracts, cools, and cleans chips before repeating the process; and post-drill reaming, in which a reamer removes a layer of material 0.1–0.2 mm thick. To increase the wear resistance of the tool, drills with a

cutting part made of hard alloys, diamond, or high-speed steel coated with titanium nitride are used [7,15,40,41].

The described phenomena and problems of implementation of the composites drilling are characterized by the term "machinability", which the author of the article [22] does not consider the exact term even in relation to the drilling of metals. Much more than in the case of metals, the number of factors affect the machinability of layered composites [15], which is usually characterized by the following criteria: 1—ease of chip removal, 2—acceptable roughness value, 3—tool life, 4—low cutting forces. That is why, although the number of articles published on the machinability of polymer glass fiber and carbon fiber composites until 2020 is about 46,000, a satisfactory model describing the dependence of these criteria on the properties of the material and the parameters of the drilling process has not been developed.

Various empirical models [10,18], approaches based on statistical analysis of variations (ANOVA) [4,42,43], and ANN models [19,20,44,45] were used to express such relationships. The ANN approach requires a large amount of experiments to generate training information, but it has clear advantages over the use of polynomial and other analytical models, the order of which, sufficient to describe the essential features of the functional relationships between the conditions and results of a complex process, can lead to overfitting, that is, to the appearance of false features.

The purpose of the presented work was to develop a rational technology for drilling holes in the root part of the composite spar for attaching the blade to the main rotor hub of the helicopter. This technology was supposed to minimize delamination at the entry and exit of the drill from the hole, thereby improving the structural integrity and reliability of the blades. Experimental studies were carried out on special samples of a thick-walled composite, in which fiberglass layers alternated with the thin titanium layers. In experiments to generate training sets for an artificial neural network, three types of drills of various geometries and nine different drilling modes were used, in each of which the cutting speeds and feed rates varied at three levels. Quantitative assessment of the magnitude of the resulting delaminations was carried out using a computer system for processing images imported from a microscope. Based on the results of preliminary experiments demonstrating the better results given by the two modified drill designs (double point angle twist drill and dagger drill) compared to the conventional geometry drill, massive process tests were carried out only with the modified geometry drills. The analysis of dependencies offered by the constructed ANNs allowed for recommending the best achievable drilling modes for each type of modified drills. The following sections are presented in the article: Materials and Tools; Experiments; Results; and Conclusions.

## 2. Materials and Tools

The object under study was a spar of a helicopter main rotor blade, made by a filament winding on a mandrel made of aluminum alloy. Unidirectional fiberglass tape VMPS 6-7 (SkyCarbon Co., Russia), impregnated with EDT-10P epoxy resin (Condor Co., Russia), was laid spirally with a winding angle varying from $\pm 45°$ at the spar root to $\pm30°$ as it approaches the tip of the blade. These and all of the following laying angles are measured relative to the longitudinal axis of the blade. Physical and mechanical properties of the glass fiber presented in Table 1 were obtained from the manufacturer, whereas the properties of the resin cured according to the current spar manufacturing technology, which are present in Table 2, were investigated using the ASTM standards and refined data processing technique [46].

In the root part of the spar, which has a rectangular cross section, 4 holes are drilled for attaching the blade to the main rotor hub and one hole for connecting to the blade pitch control rod (see Figure 1a). The thickness of the walls subjected to drilling is 36 mm, of which a set of seven reinforcement packs has a thickness of 21 mm (see Figure 1b). The reinforcement packs consist of seven identical layers, each of which contains a two-layered cross-ply laminate [0°, 90°], stacked with a layer of titanium ribbon 0.1 mm thick. The

post-curing properties of the balanced cross-ply laminates $[-45°, +45°]_6$ are presented in Table 3, whereas the manufacturer data of titanium alloy VT1-0 (RosTechCom Trade Co., Russia) are shown in Table 4.

**Table 1.** Physical and mechanical properties of E-glass fiber.

| Property | Value |
|---|---|
| Mass density, kg/m$^3$ | 2400 ... 2550 |
| Fiberglass diameter, µm | 6 |
| Tensile strength, ultimate, MPa, | 2500 ... 3200 |
| Elastic modulus, GPa | 72 ... 78 |
| Poisson ratio | 0.185 ... 0.215 |
| Shear modulus, GPa | 28 ... 30 |
| CTE, linear, µm/m·°C | 5.2 ... 5.5 |
| Specific heat capacity, J/kg·°C | 960 ... 1050 |
| Thermal conductivity, W/m·°C | 1.2 ... 1.25 |

**Table 2.** Physical and mechanical properties of post-cured epoxy resin.

| Property | Value |
|---|---|
| Mass density, kg/m$^3$ | 1200 |
| Elastic modulus, GPa | 2.4 ... 2.7 |
| Poisson ratio | 0.353 ... 0.375 |
| Softening point, °C | 150 |
| CTE, linear, µm/m·°C | 25 ... 30 |
| Linear mold shrinkage, µm/cm | 12 ... 15 |
| Specific heat capacity, J/kg·°C | 960 ... 1050 |
| Thermal conductivity, W/m·°C | 0.32 ... 0.39 |

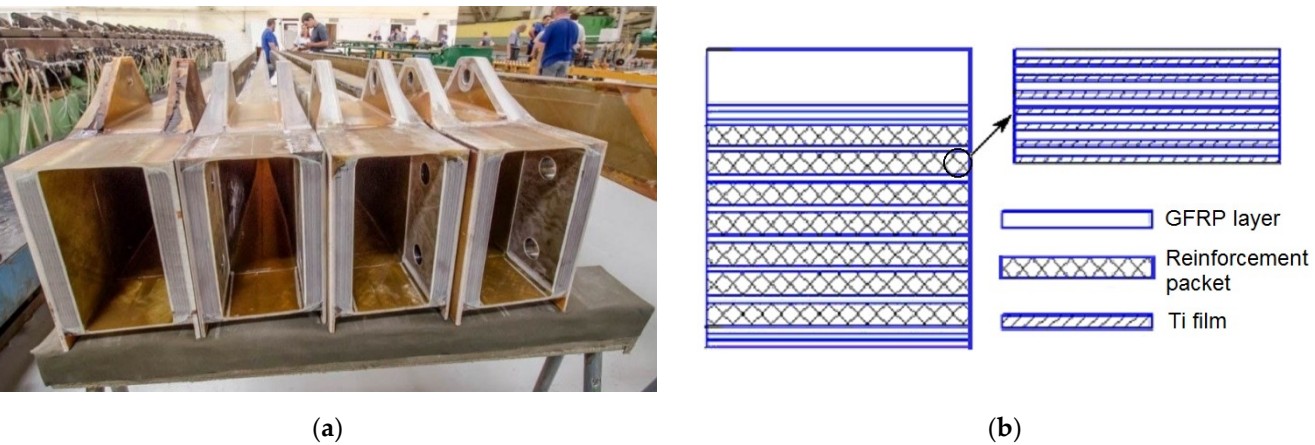

(**a**)                    (**b**)

**Figure 1.** View of the root part of the spar before and after drilling the mounting holes (**a**); (**b**) layered structure of the drilled wall.

**Table 3.** Mechanical properties of cross-ply GFRP laminate in the reinforcement packets.

| Property | Value |
|---|---|
| Mass density, kg/m$^3$ | 1800 |
| Fiber volume fraction | 0.52 ... 0.54 |
| Maximum elastic modulus ($\pm45°$), GPa | 24 ... 28 |
| Minimum elastic modulus ($0°; 180°$), GPa | 8.5 ... 10.0 |
| Poisson ratio ($\pm45°$) | 0.72 ... 0.76 |
| Shear modulus ($\pm45°$), GPa | 18 ... 22 |

**Table 4.** Physical and mechanical properties of Ti layers VT1-0.

| Property | Value |
|---|---|
| Mass density, kg/m$^3$ | 4600 . . . 4750 |
| Titanium ribbon thickness, mm | 0.1 |
| Hardness, Brinell | 170 . . . 200 |
| Tensile strength, yield, MPa | 350 . . . 380 |
| Shear strength, MPa | 380 . . . 400 |
| Elastic modulus, GPa | 102 . . . 110 |
| Poisson ratio | 0.32 . . . 0.34 |
| Shear modulus, GPa | 39 . . . 42 |
| CTE, linear, μm/m °C | 10 . . . 12 |
| Specific heat capacity, J/kg °C | 510 . . . 540 |
| Thermal conductivity, W/m °C | 16 . . . 18 |

The complex layered structure of the root part of the spar and the presence of titanium reinforcements make the production of high quality holes difficult due to the contradiction between the requirements for processing metals and fiberglass. Typical defects of hole drilling are fraying the edges of the hole and delamination, cracking, melting of the polymer matrix, and microcracks between the fiber and the matrix near the hole surface, which weakens the structural strength of the product and increases moisture absorption.

Despite a large number of studies of the effect of drill geometry on the quality of holes obtained in layered polymer composites [3,4,13,15,21,26,40,47,48], until now, specialists have not been able to develop universal recommendations for the rational choice of tools. Obviously, this is due to the variety of properties of processed materials and the types of drills. We studied the drilling process with three types of drills: conventional drill (Type 1), double-point angle drill (Type 2), and dagger drill (Type 3)—all coated with titanium nitride. The geometry of the modified drills is shown in Figure 2a. In preliminary experiments, the drilling torque, vibration activity of the drilling process, the average temperature in the hole immediately after drilling and the character of the chip formation were studied. Drilling operations were performed on a CNC (computer numerical control)-controlled high-precision machining center MAZAK FJV 35/80 JI (Yamazaki Mazak Corp., Japan). Measurements of the drilling torque were made using a specially designed device, on the plate of which the drilled specimen was fixed. The turning of the plate under the torque twisted the angular spring, bending the cantilever beam with strain gauges installed on it. The signal from the bridge was fed to an analog-to-digital (ADC) converter and further processed on a computer in the MATLAB (Mathworks Inc., Natick, MA, USA) environment. The average temperature at the surface of the hole was measured with an MS6530 (Precision Mastech Enterprises, Hong Kong, China) infrared pyrometer.

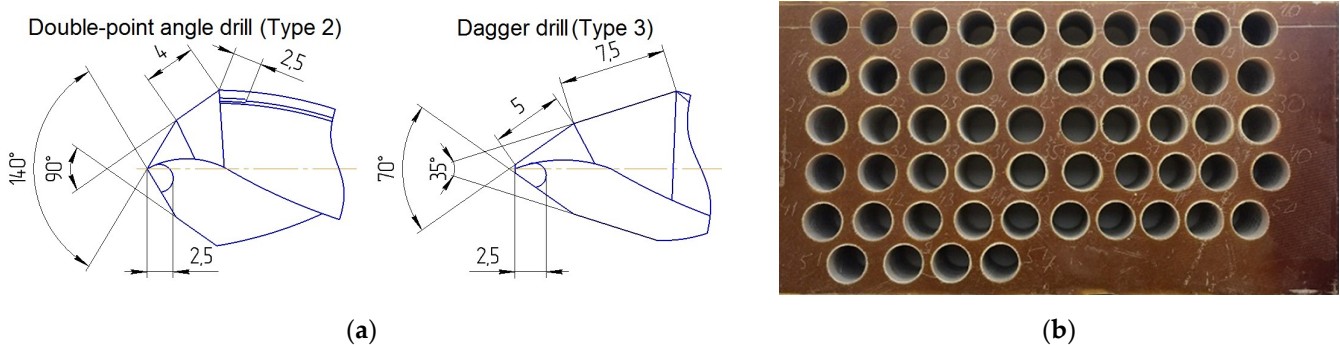

(**a**)            (**b**)

**Figure 2.** Geometry of modified drills (**a**); (**b**) view of the sample with drilled holes.

Observations of the chip formation process, which was carried out using a Canon PowerShot SX70 (Canon Inc., Tokyo, Japan) digital camera, showed that in the grooves of a conventional drill there are dense chip packets that do not crumble without external

influence. After the drill is removed from the hole, the dimensions of the chip packs increase several times under the action of expanding forces, which indicates that the chip packs are elastically clamped in the hole and press on its surface. When using a Type 2 drill, there is practically no chip packaging. Drill type 3 showed an intermediate result—chip packeting arises in certain processing modes, and when this drill is removed from a hole, the chip pack falls off. The geometry of Type 2 and Type 3 drills helps to remove the chip layer from the machined surface and direct it along the bottom of the chip groove, increasing the speed of chip movement along the drill axis. In addition, at the point of intersection of the main cutting edges of the drill, chip "break" occurs, which facilitates its transportation along the chip grooves and prevents the formation of dense chip packages. This conclusion is confirmed by measurements of the average temperature on the surface of the newly machined holes. The highest temperature value ($65 \pm 3$ °C) is achieved when using a Type 1 drill, the smallest ($52 \pm 2$ °C)—for Type 2 drills, and an intermediate temperature value ($58 \pm 3$ °C) as a result of drilling with a Type 3 tool. When changing the diameter of drills in the range of 12 . . . 24 mm at unvaried cutting speeds and feed rates, the character of chip formation did not change radically. The temperature values in the holes were also within the boundaries just indicated. Therefore, when processing all holes in the experiments, the drills of each geometry with same diameter of 15 mm were used. To eliminate the effect of the drilled hole on the neighboring ones and to select the safe distance between them, a Mitutoyo 468-164 Series 468 Digimatic Holtest 3-Point Internal Micrometer, 12 to 16 mm (Mitutoyo Corp., Kanagawa, Japan) ($\pm 0.01$ mm) was used to check the deviation of their cross sections from an ideal circle. The samples cut from a real spar, after drilling holes in it, is shown in Figure 2b.

The delamination measurements on the front and back sides of the drilled samples were carried out using a MahrVision MM220 (Mahr GmbH, Göttingen, Germany) measuring microscope with M2 geometry measuring software, which provides measurement, definition, and construction of 2D (X,Y) elements and various built-in calculation methods (best fit, best form, minimum circumscribing circle, maximum inscribed circle, etc.). In accordance with the generally accepted methodology (see review papers [2,15,29]), the amount of delamination was taken as half the difference between the maximum diameter $D_{max}$ of the circle circumscribed around the layered area and the nominal diameter D of the hole. In Figure 3, this value is represented by the letter d.

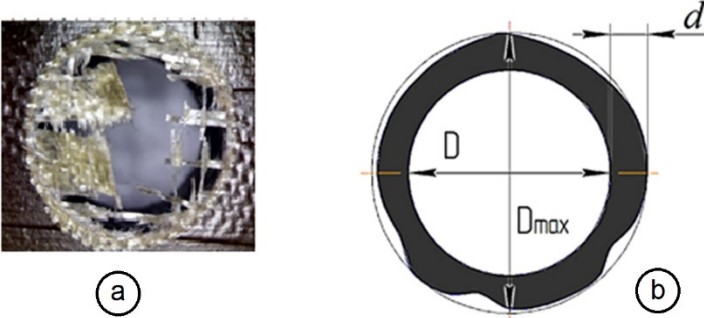

**Figure 3.** Evaluation of delamination at the entry and exit of the drill from the hole: (**a**) a shot of the exit point of the drill from the hole; (**b**) delamination measurement scheme.

## 3. Experiments

To determine the intervals of variation of the controlled parameters of the drilling process—drilling speed and feed rate, a number of preliminary experiments were performed. Despite the fact that in many works, e.g., [2,3,6,8,13,15,29,43,48], it is indicated that the amount of delamination decreases with increasing cutting speed, its upper limit was restricted due to elevated temperature, which was established experimentally. Based on the recommendations of these works and our own experience gained in the industrial

manufacture of spars, a variation interval of 7.5–19 m/min was adopted for the cutting speed, and 0.2–0.8 mm/rev for feed rate.

Experimental drilling with each of the three drills at the first stage of the study was carried out at nine points in the space of variable process parameters with a fivefold repetition of drilling in each mode. The delamination measurements were made at the drill entry into the hole and at the exit from it. Mean values, variances and confidence intervals at a confidence probability of 95% were calculated using the standard technique. The obtained discrete points for each drilling condition with each drill were fitted by 2nd degree polynomials to obtain a qualitative picture of the effect of drilling modes on the resulting delaminations. The results of preliminary experiments in the form of response surfaces, depicted together with mean values and confidence interval boundaries for each drilling mode, are shown in Figure 4. In order to use the results of our study for drills diameters within above mentioned range, meter per minute (m/min) was adopted as a unit of measurement for drilling speed, in contrast to the often-used revolutions per minute (rpm). A comparison of the response surfaces presented in this figure shows that the modified drill designs (double-point angle drill and dagger drill) demonstrate less delamination than conventional drill for all drilling modes. This inference made it possible to exclude the conventional drill from further research, which involved the use of artificial neural networks to analyze the response functions of each tool in more detail and perform massive experiments to generate training data.

When choosing the ANN architecture, the number of layers and neurons in the created neural networks, the "golden rule of modeling" was used. Namely, the model should adequately describe the properties of simulated system that are of interest to the researcher; be able to describe, possibly, a wider class of modeling phenomena; have the simplest structure, the minimum number of parameters required to describe the simulated phenomena, avoiding the appearance of false features; and have a minimum computational cost. Based on this, the advantage over recurrent, radial-basic networks was afforded to feedforward neural networks, wherein connections between the nodes do not form a cycle or loops. This choice was dictated by the following considerations: The universal approximation theorem [49,50], developed in the ANN theory, states that every continuous function that maps intervals of real numbers to some output interval of real numbers can be approximated arbitrarily closely by a multi-layer perceptron with just one hidden layer. This result holds for a wide range of activation functions, e.g., for the sigmoidal functions. A generalization of these results is provided in [51,52], where it is proven that a network of width n with a ReLU (Rectified Linear Unit) activation function can approximate a Lebesgue integrable function defined in the space of n dimensions if the network depth, i.e., number of hidden layers, is sufficient. The dependences modeled in this work certainly satisfy these conditions and do not require Lebesgue integrability, since they should not have any singularities. When analyzing the feasibility of using radial-basic networks to solve the problem, it was noted that they are the most effective for interpolation. However, if the purpose is the approximation of functions, the optimization of the weights becomes somewhat more complicated. Since the number of points in the input space is limited, there is no obvious choice for basis function centers, and the training is carried out in two stages: first, the width and centers are fixed, and then the weights are determined. In this regard, these architecture options, as well as recurrent ANNs, were rejected due to their complexity.

The use of ANN requires that the number of elements in the training sample be sufficient for network training, but the number of these elements is not strictly regulated. When solving the problem under consideration, it was experimentally found that a sample of 154 elements obtained after removing outliers (1 . . . 3 for each data set) is sufficient for reliable training of all generated networks.

According to the requirements of the neural network fitting tool (nftool) in the MAT-LAB [53] environment, the data sets were divided into three kinds of samples: one of which (90 elements) was used for training, and two parts, each with 27 elements, were used for validation and testing. Thus, in each mode, each of the drills performed drilling 16 times.

To take into account possible random deviations of drilling modes, the range of which is regulated by the technical data of the machine, the values of cutting speeds and feeds were randomized within these intervals in accordance with the normal distribution, and, together with the delamination data for each experiment, were loaded onto the workspace.

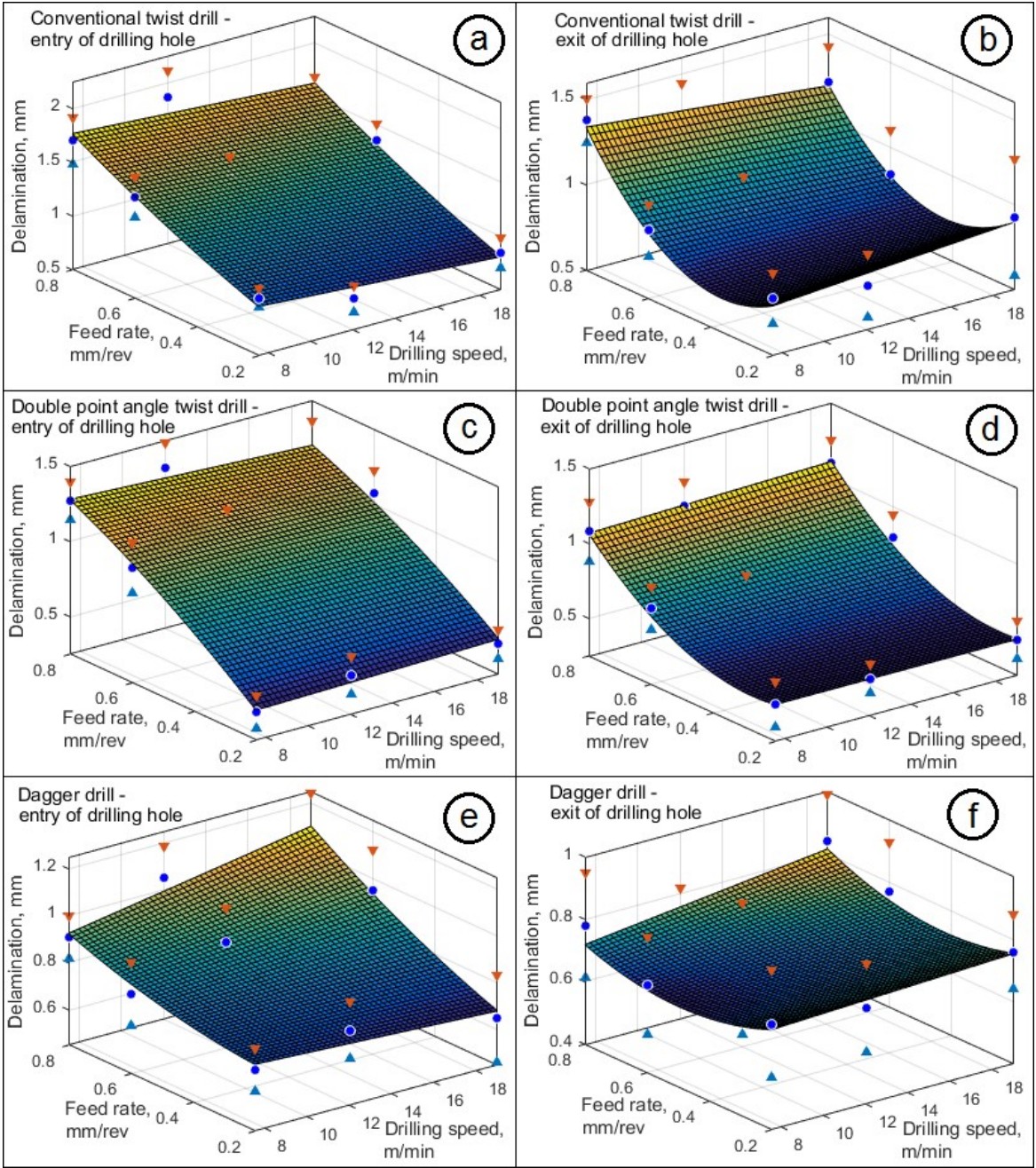

**Figure 4.** Three-dimensional plots of delamination values at the entry of drills into the hole (**a**,**c**,**e**) and at the exit from the hole (**b**,**d**,**f**) for a conventional twist drill (**a**,**b**), a double-point angle twist drill (**c**,**d**) and a dagger drill (**e**,**f**) represented as second-order polynomial-fitted response surfaces combined with the delamination mean values (●), upper (▼), and lower (▲) bounds of the confidence intervals for each process test.

The adopted ANN architecture was a feed-forward network with input layer and one hidden layer of tansig neurons followed by an output layer of linear neuron. For the studied two-point angle twist drill and dagger drill, we used three and five neurons, respectively,

in the hidden layer of the ANN. To exclude cases of underfitting or overfitting, network models with different numbers of neurons in hidden layers and different types of transfer function were studied. Each such model was repeatedly retrained with a random change in each of the sets (training, verification, and testing) and performance monitoring. It included analyzing the performance progress graph to find the point at which the validation test curve reaches a minimum, which is automatically determined by the early stop option as the training curve continues to decrease for another six epochs to avoid overfitting. Figure 2a demonstrates an example of the performance dependence vs. iterations for the ANN of three hidden layers predicting delamination in the entry of hole drilled by the tool Type 2. The final distribution of the errors for corresponding data set is present by histogram in Figure 5b. A more detailed understanding of the quality of the approximation given by this neural network is provided by the regression plots for the training set, the validation set, the test set, and all three sets combined together (see Figure 6). Very high correlation coefficients, exceeding 0.9, show that the quality of the experimental data fitting by this model is very high. Based on a similar analysis of regression plots showing the relationship between network outputs and targets, the remaining ANNs were selected with a maximum regression coefficient, but not less than 0.85. It should be noted that more complex networks, such as those used to predict delaminations using a Type 3 drill and containing five hidden neurons, require more epochs (10 . . . 15) to achieve optimum performance.

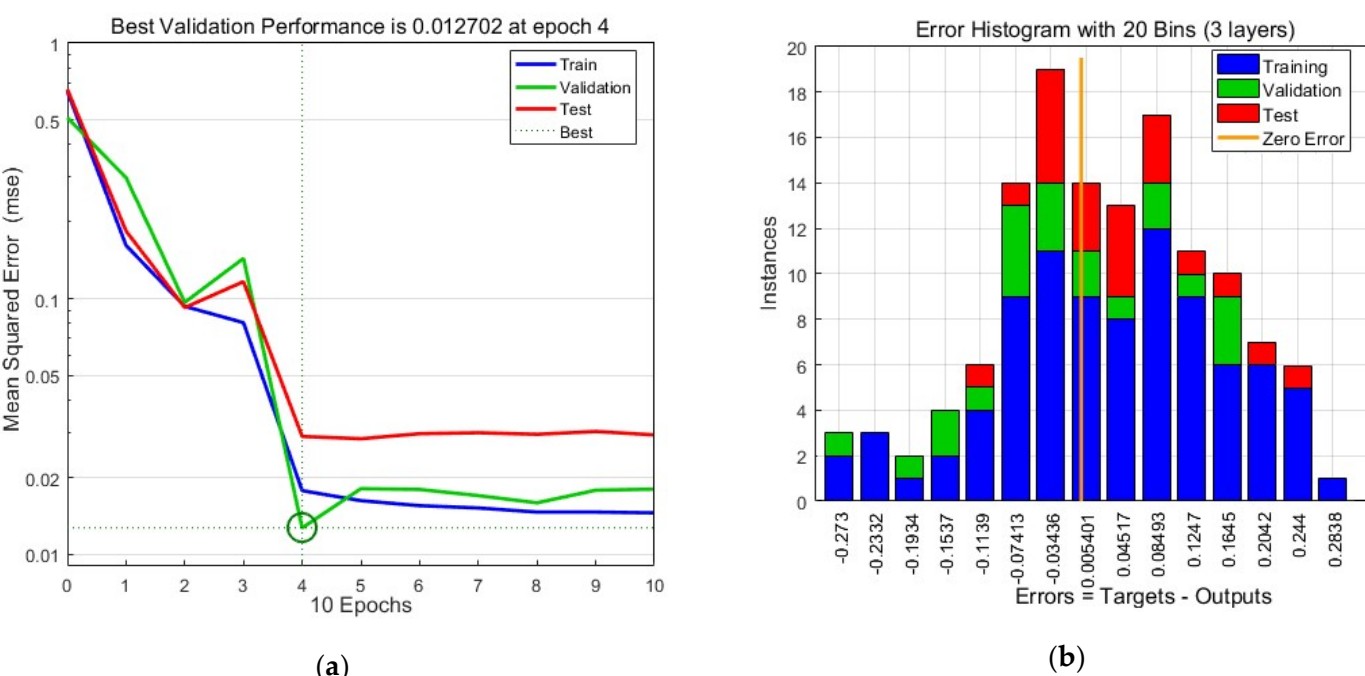

**Figure 5.** The plot, which indicates the iteration at which the validation performance reached a minimum (**a**) and histogram, which demonstrates scattering the output values the training, validation and testing sets of the delamination model for the drill Type 2 entry to the hole after four epochs (**b**).

The training of these networks was carried out with the Levenberg–Marquardt back-propagation algorithm, which showed the best quality of training. After adjustments and retraining, all adopted versions of the networks were used to create the response surfaces for the delamination, which are shown in Figures 7 and 8.

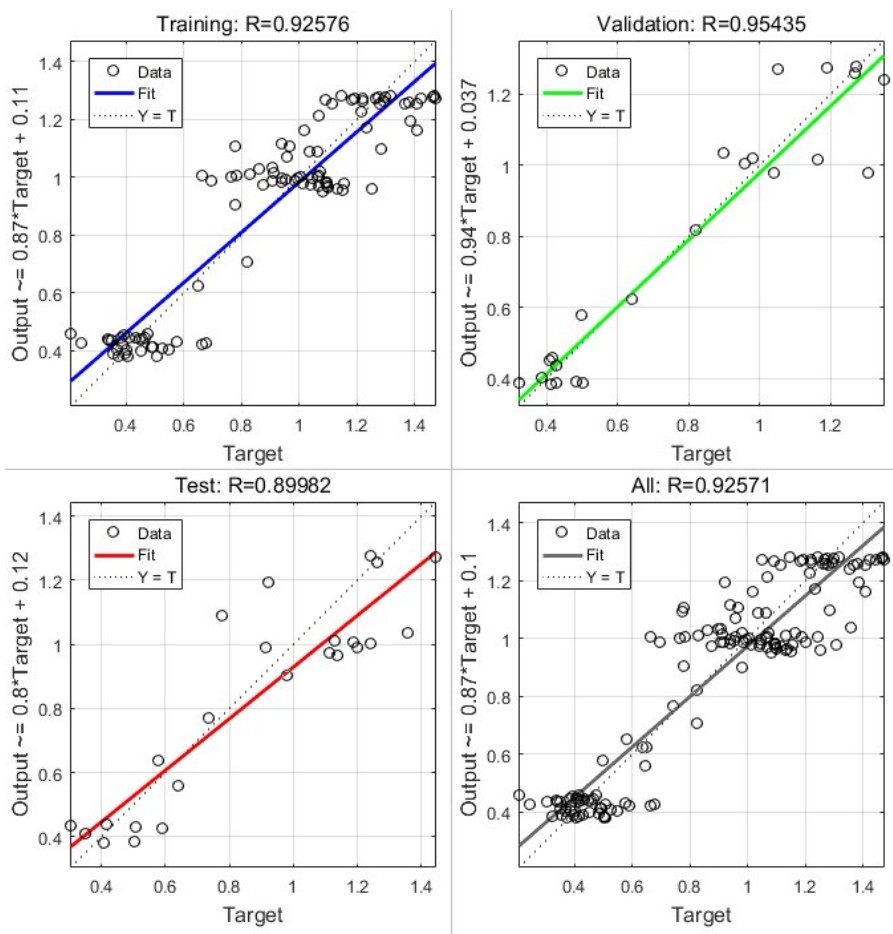

**Figure 6.** ANN-fitted delamination response functions at the entry (**top**) and exit (**bottom**) of a hole drilled with double-point angle drill is presented as 3D surfaces (**left**) and contour plots (**right**).

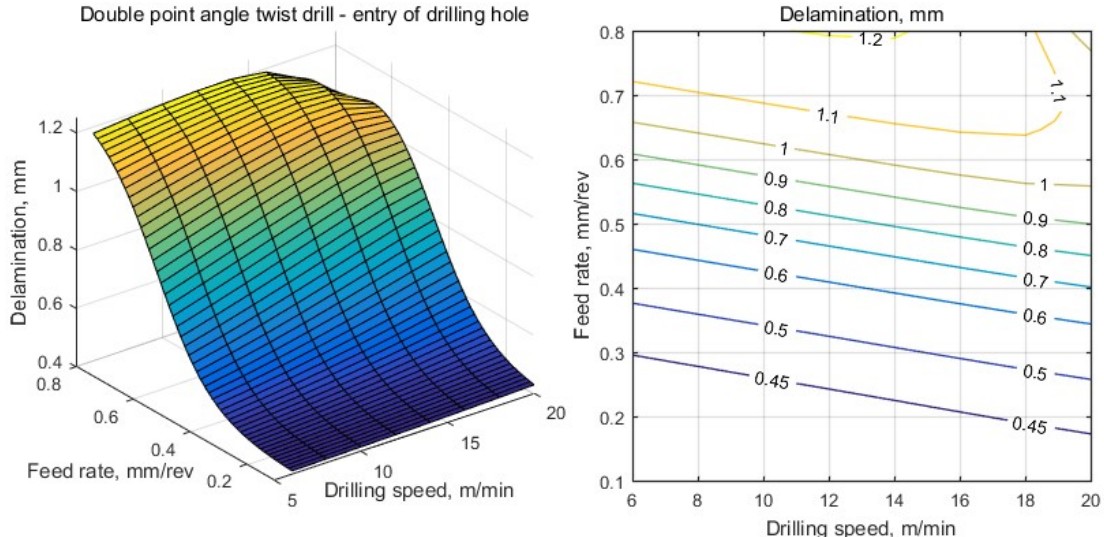

**Figure 7.** *Cont.*

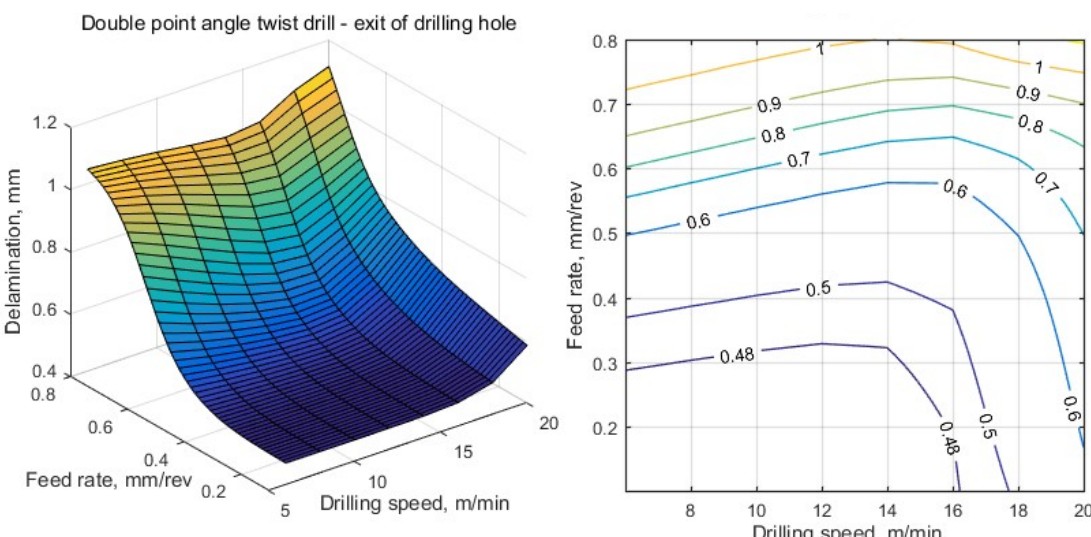

**Figure 7.** ANN-fitted delamination response functions at the entry (**top**) and exit (**bottom**) of a hole drilled with double-point angle drill is presented as 3D surfaces (**left**) and contour plots (**right**).

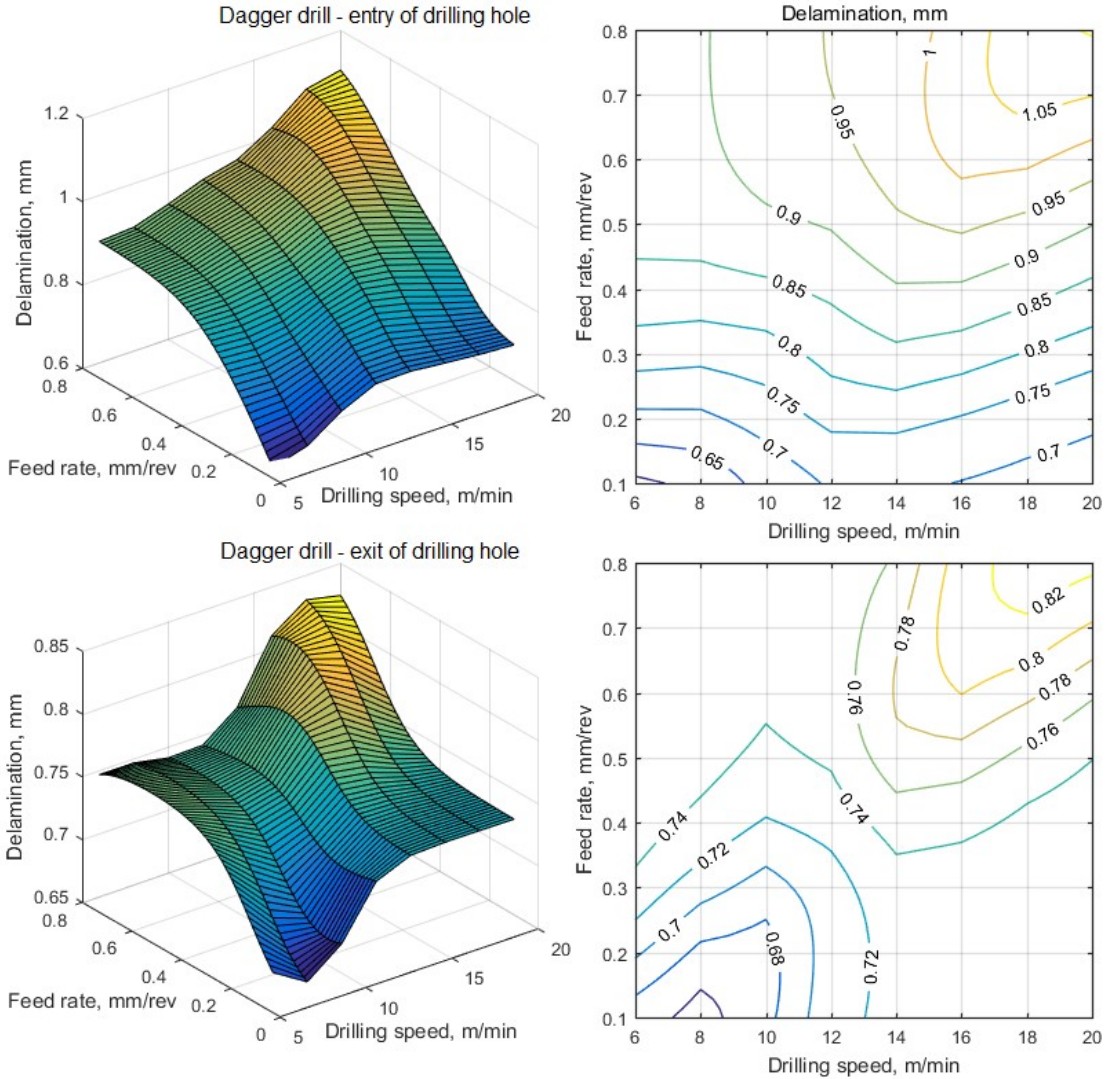

**Figure 8.** ANN-fitted delamination response functions at the entry (**top**) and exit (**bottom**) of a hole drilled with dagger drill is presented as 3D surfaces (**left**) and contour plots (**right**).

## 4. Results

For better visualization and quantification, 3D images of the response surface are shown together with the level line plots. The most important and expected trend of delamination dependences, observed both at the entry and at the exit of both drill types from the hole, is a significant increase in delamination with increasing feed.

At a relatively low feed rate, an increase in cutting speed leads to a very slight increase in delamination at the drill entry. For a two-point angle twist drill (Type 2), the slight increase in delamination at the drill entry is very small and within the measurement accuracy, while for a dagger drill (Type 3), delamination begins to increase sharply already at speeds of ~7 m/min. This result can be explained by the lengthened taper part of the Type 3 drill, which leads to a longer removal of material layers from the moment the drill enters to reaching a given diameter and, consequently, a longer warm-up of the material adjacent to the hole.

At the exit of a Type 2 drill, a slight increase in delamination with increasing cutting speed starts from 15 m/min. This is explained by the fact that this growth is due to a gradual increase in temperature in the body of the composite as the drill moves towards the exit from the hole. In addition, the composite layer in front of the moving drill continuously becomes thinner, its mass and heat capacity decrease, and due to the low thermal conductivity of fiberglass, the main heat is released in the layer in front of the drill, i.e., hole exit. However, the dependence of the amount of delamination at the exit for Type 3 drills on cutting speed is much stronger. Its sharp growth starts from about 7 m/min, as at the entrance, and leads to an increase in delaminations from 0.65 mm to 0.84 mm, i.e., by 30%. This effect can also be explained by the presence of an elongated conical part of the dagger drill, in which the movement of chips is difficult. In addition, the path of the drill from the entry to the completion of drilling in this case is significantly longer than when using the double-point angle drill.

It is convenient to make a quantitative comparison of the characteristics of delaminations formed by both types of drills using maps of level lines. They show that the area of minimal delamination, both at the entry and exit of the dagger drill, is much smaller than when using the double-point angle drill. This area for the Type 3 drill is limited by feed rates not exceeding 0.2 mm/rev and speeds up to 10 m/min. The value of this area is much smaller than the similar area for the double-point angle Type 2 drill. This means that the drilling technology using dagger drills is much less reliable, and the value of the minimum achievable delamination is approximately 1.5 times greater: $0.68/0.45 \approx 0.7/0.48 \approx 1.5$ (see Figure 9a,b). Thus, the use of the double-point angle drill is preferred. Unfortunately, when drilling the structure under consideration, it is very difficult to use a support tightly pressed against the back side of the wall being drilled, since the strict requirement of alignment forces drilling of both coaxial holes in one drill pass, and the closed box-shaped design of the spar makes it difficult to install any supports.

The dependences obtained qualitatively fully correspond to the results of the studies cited above [10–13,17,18,20,29,42,54] both in terms of the nature and value of the delamination at the entry and exit of drills, and in terms of the dependence of these defects on the drill geometry and drilling modes. However, a complete quantitative comparison of the results obtained in the work with the results of the cited studies is impossible and illegal due to the significant difference in the properties of the processed materials. A large and ever-expanding variety of composite materials, composite/metal layered structures required for machining significantly complicates the development of a system for classifying and ranking the machinability of such structures [2,3,15,29], an analogue of which is currently developed for metal alloys and is being continuously improved [22,55]. The creation of such a system, which requires the efforts of specialists from many countries, will require the implementation of huge amounts of research using not only the most modern scientific methods, but also means for detecting and assessing the degree of defects, suitable for real production conditions. This circumstance is the reason for the ever-increasing growth in the number of publications on the subject under consideration. In addition, methods for

quantifying the causes and severity of defects using neural network technologies are an inevitable and indispensable component in the development of an optimal technology for machining composite structures.

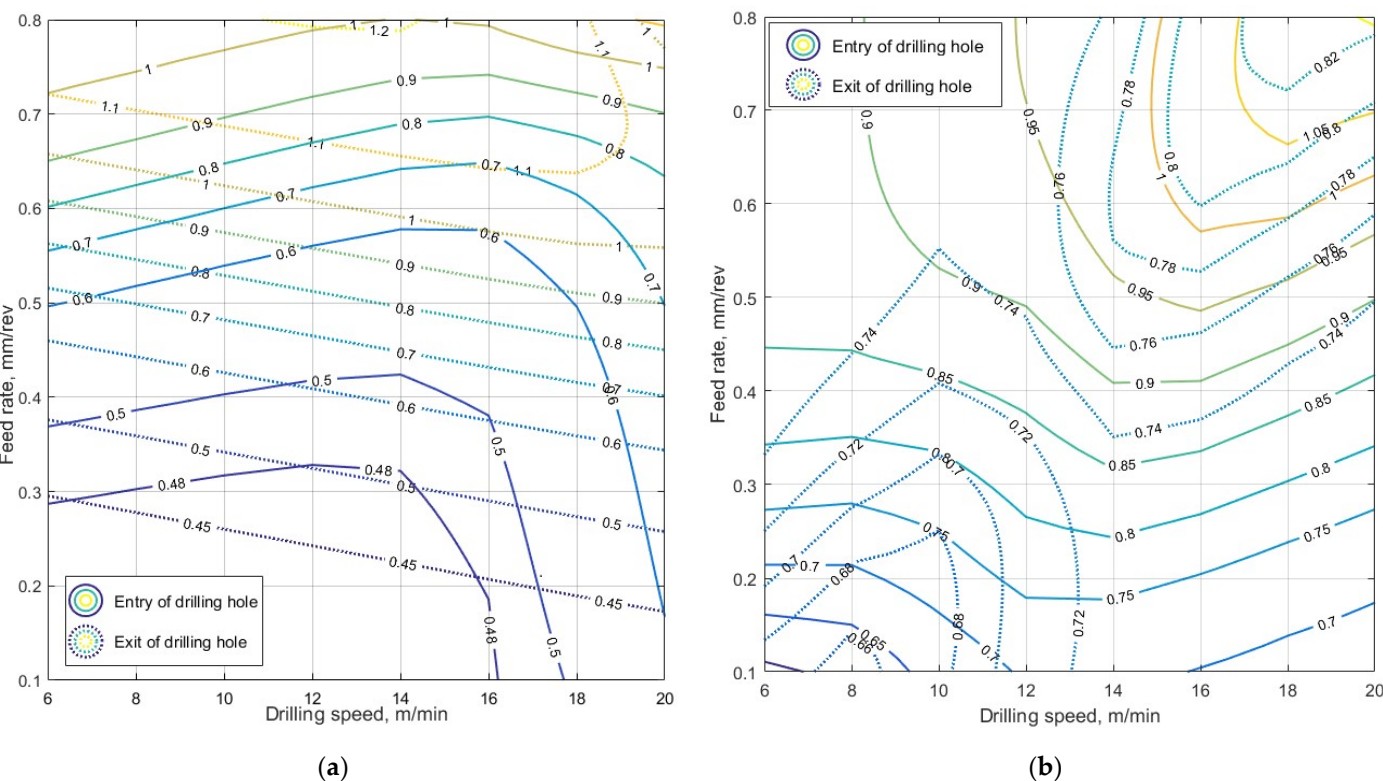

**Figure 9.** Superimposed level lines maps of predicted delaminations at the entry (solid lines) and at the exit (dashed lines) from the hole of the double-point angle drill (**a**) and the dagger drill (**b**).

## 5. Conclusions

In this article, the means of reducing delaminations in the holes of the high loaded root part of the helicopter composite blade spar, by which the blade is attached to the main rotor hub, were studied. Drilling of a thick-walled polymer composite with interlayer reinforcing titanium plates was carried out by three designs of drills of various geometries with a variation in drilling modes—cutting speed and feed. Based on the analysis of a series of preliminary experiments, a plan was developed for conducting mass experiments to reconstruct the dependences of the amount of delaminations on drilling modes in the form of fitting function. The results of the study of emerging delaminations, recorded by a computer system for analyzing images of a microscope, were used to form training sets of neural networks designed to model the dependence of the main quality criterion—the maximum delaminations at the entry and exit of a hole. The results of the study, presented in the form of response surfaces of these criteria to drilling modes and the superimposed contour lines in the speed/feed coordinates, demonstrated the best results obtained using double-point angle drills that ensure the achievement of minimal delamination, both at the entry and at the exit of the drill from the hole. It has been established that the optimal drilling results—delamination values up to 0.48 mm at the entry and up to 0.46 mm at the exit—can be obtained in the cutting speed range of 6 . . . 15 m/min and feed rates of 0.1 . . . 0.25 mm/rev. Experimental observations of chip formation and temperature in the area adjacent to the drilled hole have established the possibility of using the developed recommendations for the rational choice of cutting speeds and feed rates in a certain range of drilling diameters. The methodology based on the use of ANN has made it possible, with a limited number of technological tests, to obtain reasonable estimates of the achievable

quality and performance objectives limited in the production of composite structures with increased requirements for structural strength.

**Author Contributions:** Conceptualization, S.S. and E.C.; methodology, S.S. and I.Z.; software, I.Z.; validation, E.C. and I.Z.; formal analysis, S.S.; investigation, E.C. and S.C.; resources, S.C.; data curation, N.S.; writing—original draft preparation, E.C. and S.C.; writing—review and editing, S.S.; visualization, N.S.; supervision, S.S.; project administration, E.C. All authors have read and agreed to the published version of the manuscript.

**Funding:** This research received no external funding.

**Conflicts of Interest:** The authors declare no conflict of interest.

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
