# Peer review of "ANN-Based Estimation of the Defect Severity in the Drilling of GFRP/Ti Multilayered Composite Structure"

_jcs, doi:10.3390/jcs6120370_

Round 1
Reviewer 1 Report
Please, see the attachment

Author Response
- Line 13. “GFRP-Ti”, “ANN”. Please, clarify the meaning of the acronym when the first time used in the Abstract. (see lines 13, 19)
- The Abstract needs significant improvement. It has a lot of missing information regarding the research results. The findings are somewhat generic. Results (in terms of numbers and their comparison) are not specified at all. Abstract could be more informative by providing results in terms of the numbers. Add some key values from results and highlight the novelty of this work clearly. (Abstract text has been reorganized)
- Used keywords are inappropriate. (Some keywords were changed)
- Line 54. “GFRP (CFRP)” – see my first comment. (See also lines 55, 88 and 174-175)
- Line 88. “Therefore, each new publication on this topic is useful at least for practice.” Please, use concise English and delete unnecessary sentences. It just enlarges the manuscript and makes it harder to process.
(This sentence removed. The text has been corrected.)
- Selected references are quite old, which from the one point of view is good, since the Authors cited necessary references to define a research problem, while from the other hand, lack of recent references may indicate an insufficiently performed literature review. Try to add some more recent and up-to-date research papers, especially from 2021-2022. (Many new and recommended references have been added, see References list)
- In order to emphasize the importance of the subject the Authors are encouraged to mention manufacturing induced defects arising in the FRP composites on the production stage which will in turn affect the performance of the structures during drilling. Please, refer to: (The introduction has been significantly revised in accordance with your comments. Please, look at lines 53-114)
- What tools were used for the drilling. Please, specify it in the text. (see lines 172-174)
- Lines 139-140. Please, specify the name of the system, its manufacturer and country of origin. (see lines 178-179, 184-185)
- According to Fig. 2b, all the holes are located quite close one to each other. Therefore, the residual stresses from one hole might affect defects occurring during the drilling of the nearest holes. Therefore, the results of this study will be also affected. Please, elaborate on this point in the manuscript. (This problem is considered in lines 202-205)
- Line 182. “the training set was divided into three kinds of samples”. Not the training set, bud the obtained dataset was divided. It’s incorrect to say that training set was divided into training, testing and validation. Please, correct.(Thank you, my erratum is removed, see line 277)
- What was the reason to use this particular architecture and not the other? (See lines 256-270)
- Please, demonstrate the effect of the activation function choice on the accuracy of your NN. Try, for example, ReLU, tanh etc. Compare the obtained results with the result of your model based on sigmoid function. (see lines 289-297)
- Line 191. “networks received satisfactory performance”. What were the evaluation metrics? What is the accuracy of the model? None of this has been presented and this is serious flaw of this manuscript. I assume if the name of the paper has “ANN”, then much more attention should be paid to the NN part itself. Now this is limited to just several lines. This part must be significantly improved. (This part was fully reorganized. See lines 292-297)
- Are you sure that the developed model has no underfitting/overfitting? Please, examine it within the manuscript. (See lines 289-293 and 298-302)
- The Authors are encouraged to bring additional paragraph discussing possibilities for future studies, showing future research directions to the scholars. Besides, deeper analysis regarding the possible practical applications of results revealed in this study is needed. (Suggested part has been inserted, see lines 367-378)
- Please, compare your results with published papers in the related field and explain the similarity and differences between your results and theirs. (See lines 360-365)
- Please, mention the limitation of your study. (See answer 17)

Reviewer 2 Report
The paper presents an interesting approach based on the ANN Based Estimation of the Defects Severity at the Drilling of GFRP/Ti Multilayered Composite Structure. However, the innovation of the current research work should be further highlighted and emphasized. At the same time, the authors should consider the following comments to greatly improve the quality of the paper.
1. In the abstract, kindly introduce the problem in the initial lines.
2. The introduction needs to be improved by relating to the mechanics of the studied materials and their mechanical characteristics. The references to be included are: 10.1016/j.polymertesting.2017.09.009, 10.1016/j.compstruct.2021.114698, 10.1177/0731684417727143, 10.1002/app.46770, 10.1016/j.porgcoat.2022.107015.
3. Kindly add a table that describes the main physical and chemical properties of the raw materials used in this study.
4. What is the type of fiber glass that was used as part of rotor blade? The orientation of fibers, their diameter and volume fraction are very significant information that must be provided.
5. The authors are advised to provide the drilling data when using different diameters of the drills. The chip formation is expected to vary accordingly.
6. Were the preparation methods and delamination measurements described by the authors come in accordance with a certain standard or do they follow previous procedures?
7. Why do the authors used the feed-forward network to represent the ANN algorithm? Have you tried using the other modules and making accuracy comparisons between the models? This part is missing in the work and needs extensive justification to the approach you decided to use.
8. The conclusion needs to be modified to summarize the research outcomes in short statements with clear observations.
Author Response
- In the abstract, kindly introduce the problem in the initial lines. (Abstract text has been reorganized)
- The introduction needs to be improved by relating to the mechanics of the studied materials and their mechanical characteristics. The references to be included are: 10.1016/j.polymertesting.2017.09.009, 10.1016/j.compstruct.2021.114698, 10.1177/0731684417727143, 10.1002/app.46770, 10.1016/j.porgcoat.2022.107015. (The introduction has been significantly revised in accordance with your comments, recommended references have been added)
- Kindly add a table that describes the main physical and chemical properties of the raw materials used in this study. (All recommended tables with the accessible data have been added)
- What is the type of fiber glass that was used as part of rotor blade? The orientation of fibers, their diameter and volume fraction are very significant information that must be provided. (See Tables 1-4, and lines 141-155)
- The authors are advised to provide the drilling data when using different diameters of the drills. The chip formation is expected to vary accordingly. (see lines 192-198)
- Were the preparation methods and delamination measurements described by the authors come in accordance with a certain standard or do they follow previous procedures? (There is no generally accepted standard for the determination of delaminations in laminated composites, and the technique used is generally accepted. See lines 207-209)
- Why do the authors used the feed-forward network to represent the ANN algorithm? Have you tried using the other modules and making accuracy comparisons between the models? This part is missing in the work and needs extensive justification to the approach you decided to use. (see lines 256-270)
- The conclusion needs to be modified to summarize the research outcomes in short statements with clear observations. (The text of the conclusion has been changed.)

Round 2
Reviewer 1 Report
please, see the attachment

Author Response
Dear reviewer! Thank you very much for your suggestions. We hope that our responses and appropriate corrections to the text will improve the readability and clarity of the article.
Sincerely, the authors

Reviewer 2 Report
The paper can be accepted.
Author Response
Dear reviewer! Thanks a lot for your comments and suggestions. They have been taken into account and the corresponding corrections have been made to the text.
Sincerely, authors
Round 3
Reviewer 1 Report
The revision is reasonable, paper can be accepted